# Optimization of the Navigated TMS Mapping Algorithm for Accurate Estimation of Cortical Muscle Representation Characteristics

**DOI:** 10.3390/brainsci9040088

**Published:** 2019-04-19

**Authors:** Dmitry O. Sinitsyn, Andrey Yu. Chernyavskiy, Alexandra G. Poydasheva, Ilya S. Bakulin, Natalia A. Suponeva, Michael A. Piradov

**Affiliations:** 1Department of Neurorehabilitation and Physiotherapy, Research Center of Neurology, 125367 Moscow, Russia; andrey.chernyavskiy@gmail.com (A.Yu.C.); alexandra.poydasheva@gmail.com (A.G.P.); bakulinilya@gmail.com (I.S.B.); nasu2709@mail.ru (N.A.S.); mpi711@gmail.com (M.A.P.); 2Quantum Computer Physics Laboratory, Valiev Institute of Physics and Technology of Russian Academy of Sciences, 117218 Moscow, Russia

**Keywords:** navigated transcranial magnetic stimulation, TMS motor mapping, cortical muscle representation, bootstrapping, variability, accuracy

## Abstract

Navigated transcranial magnetic stimulation (nTMS) mapping of cortical muscle representations allows noninvasive assessment of the state of a healthy or diseased motor system, and monitoring changes over time. These applications are hampered by the heterogeneity of existing mapping algorithms and the lack of detailed information about their accuracy. We aimed to find an optimal motor evoked potential (MEP) sampling scheme in the grid-based mapping algorithm in terms of the accuracy of muscle representation parameters. The abductor pollicis brevis (APB) muscles of eight healthy subjects were mapped three times on consecutive days using a seven-by-seven grid with ten stimuli per cell. The effect of the MEP variability on the parameter accuracy was assessed using bootstrapping. The accuracy of representation parameters increased with the number of stimuli without saturation up to at least ten stimuli per cell. The detailed sampling showed that the between-session representation area changes in the absence of interventions were significantly larger than the within-session fluctuations and thus could not be explained solely by the trial-to-trial variability of MEPs. The results demonstrate that the number of stimuli has no universally optimal value and must be chosen by balancing the accuracy requirements with the mapping time constraints in a given problem.

## 1. Introduction

Mapping cortical motor representations of muscles using navigated transcranial magnetic stimulation (nTMS) is a valuable noninvasive method providing information about the motor system that is useful for research and clinical purposes [1,2,3]. Its ability to localize motor eloquent cortical areas has found successful applications in preoperative planning [4,5]. Additionally, a growing body of literature is concerned with the use of nTMS mapping for assessing the state of the motor system and its plastic changes during learning of new skills [6,7,8,9], in neurological diseases, such as stroke [10], dystonia [11], spinal cord injury [12,13], amyotrophic lateral sclerosis [14], as well as during the course of treatment [15]. For identifying the possibly subtle differences in motor maps, it is essential to make the method precise and reliable. Meanwhile, the high variability of motor evoked potentials (MEPs), on which the TMS-maps are based, makes the accurate estimation of representation parameters challenging [16,17,18].

The interpretation of the results of TMS mapping is complicated by the lack of a standard protocol, and the existence of a wide variety of approaches to the mapping procedure, the selection of the studied muscle representation parameters and methods of their calculation [10,19]. One of the most frequently used approaches is based on a predefined grid of cortical points with application of a fixed number of stimuli at each point [20,21]. The studies using this method are heterogeneous in terms of the number of grid cells, their size and the number of stimuli per cell [10,21,22,23,24,25]. Given the high variability of MEPs, the number of stimuli per cell is an important factor influencing the accuracy of the representation parameters [20,26].

The reproducibility of muscle representation parameters and their stability in the absence of interventions is one of the key aspects for the application of navigated TMS motor mapping for research and clinical purposes [16,27]. The studies conducted to date have obtained divergent results with regards to the intraclass correlation coefficient (ICC) for various parameters of cortical representations, ranging from 0.36 to 0.89 [16,24,27]. A number of approaches for reducing the variability of muscle representation parameters have been proposed, such as neuronavigation by an individual structural magnetic resonance imaging (MRI) for improving the repeatability of coil placement and orientation [28,29] and taking into account the individual topography and morphology of the cerebral cortex [30,31,32]. Another promising research direction is the brain-state dependent stimulation based on combining electroencephalography (EEG) and TMS in real time to align the stimulus times with EEG features, such as the μ-rhythm phase [33].

A general approach to dealing with the trial-to-trial variability of MEPs is averaging multiple measurements [34,35]. In agreement with probability theory, the accuracy of some muscle representation parameters has been reported to increase with the number of stimuli used during mapping [20]. However, comprehensive knowledge of this dependence for all the common parameters is lacking, and it is unknown whether the increase in the accuracy saturates (reaches a plateau) after a certain number of stimuli. This is important for estimating the payoff in the quality of the data that a researcher obtains from investing the subject’s and operator’s time and effort into the detailed mapping of muscle representations.

Another open question regarding the averaging approach is whether it can reduce to an arbitrary degree the session-to-session variability of muscle representation parameters in the absence of interventions. Averaging makes the parameters closer to their exact mean (expected) values in a given session, and these values will not necessarily be the same in a different session. Thus, it is important to test whether the variations of muscle representation parameters between sessions can be fully explained by the trial-to-trial MEP variability within a session, and can thus be controlled by sufficient sampling of MEPs. An alternative scenario is the existence of systematic between-session changes of the MEP probability distributions, which cannot be influenced by the sampling scheme.

The existing data analysis methods in TMS mapping differ in their definitions of muscle representation parameters. The area of a representation mapped using the grid-based method has been defined as the total area of the cells with at least one suprathreshold MEP out of three stimuli [22], at least five out of ten [23], six out of ten [21,24], or two out of six [25] suprathreshold MEPs. Several studies have studied the area in which an interpolated mean amplitude function exceeds some threshold, with varying interpolation methods and thresholds [17,36,37]. Recently, a more advanced minimum-norm estimation procedure has been proposed [38]. There is a need for research comparing the statistical properties of these definitions of the representation area. This can help develop guidelines for selecting an appropriate definition, possibly depending on the particular TMS mapping application.

The purpose of the present study was to determine the influence of TMS mapping and data processing algorithms on the accuracy of estimating muscle representation parameters. Using a grid-based mapping approach, we studied the effect of MEP sampling, i.e., the size of the stimulation grid and the number of stimuli per cell, on the within-session accuracy and between-session variation of the muscle representation characteristics. We tested whether the between-session parameter changes could be explained by the within-session MEP variability. Additionally, we investigated the impact of the data analysis methods by comparing several alternative definitions of the representation area, weighted area and center of gravity (COG), in terms of their estimation accuracy. The results can be applied to choosing an appropriate TMS mapping algorithm for a given research or clinical problem, by finding a compromise between the accuracy requirements and mapping time constraints.

## 2. Materials and Methods

### 2.1. Subjects and the nTMS Mapping Procedure

For all subjects, an MRI was acquired in the T1 multiplanar reconstruction regime on a 3T Siemens MAGNETOM Verio clinical scanner (Siemens, Erlangen, Germany). A 3D brain surface was reconstructed based on the T1-weighted structural MR images and used in the Nexstim navigated TMS software (NBS Eximia Nexstim 3.2.2, Nexstim, Helsinki, Finland) for visualizing the brain morphology during mapping.

The navigated TMS mapping was performed using the NBS eXimia Nexstim stimulator (Nexstim, Helsinki, Finland). We used a figure-of-eight biphasic coil with a diameter of 50 mm to deliver stimuli with a 280 µs duration. The pulses induced a posterior-anterior followed by an anterior-posterior current flow in the brain. Biphasic pulses were used because they yield the lowest motor thresholds [39,40], which mitigates the problem of coil heating. The maximum value of the estimated induced electric field in the cortex was 199 V/m. The electromyographic (EMG) activity of the studied muscles was recorded using skin pre-gelled disposable electrodes (Neurosoft, Russia). A suprathreshold MEP was defined as an EMG response having a peak-to-peak amplitude greater than or equal to 50 µV in the interval from 15 to 30 ms after the stimulus.

The hot spot was identified by stimulating the hand knob and the adjacent areas by no less than 20 stimuli at an intensity sufficient for inducing MEPs with amplitudes of 100–500 µV. The point with the maximal MEP amplitude was considered the hot spot. This location was used for measuring the individual resting motor threshold (RMT) defined as the minimum intensity of stimulation for which five out of ten stimuli produced suprathreshold MEPs. The stimulation intensity during the mapping was set to 110% of the RMT. The inter-stimulus interval was greater than two seconds.

Data from two experiments were employed for answering different research questions. The first dataset was recorded previously for different purposes. It was used here to determine an optimal size of the stimulation point grid for the second (main) experiment. The dataset contained 121 TMS maps for the dominant abductor pollicis brevis (APB), extensor digitorum communis (EDC) and flexor digitorum superficialis (FDS) muscles of 33 healthy subjects (21 women, median age 27, age quartiles 25, 31; nine subjects were left-handed according to the Edinburgh handedness inventory [41]). In this experiment, the locations and sequence of the stimulation points were determined individually (without a grid), taking into account the responses obtained at previous points. Each point was stimulated once, and the mapping progressed in a given direction until obtaining two points without suprathreshold MEPs.

The second (main) experiment aimed at investigating the relationship between the MEP sampling scheme in the grid-based mapping algorithm and the representation parameter accuracy. The APB muscle was chosen because of its frequent use in TMS mapping due to its relatively large cortical representations and low baseline EMG activity [42]. The cortical representations of the right APB muscle in 8 healthy volunteers (3 women, median age 28, age quartiles 24, 29, all right-handed according to the Edinburgh handedness inventory [41]) were mapped three times on consecutive days. We used a stimulation point grid consisting of 7 × 7 square cells with a side of 7.63 mm (at the peeling depth of 20 mm), centered at the hotspot. The cells were defined with the help of the grid tool in the Nexstim stimulator software (NBS Eximia Nexstim 3.2.2, Nexstim, Helsinki, Finland). Ten rounds of stimulation were performed, and in each round, a single stimulus was applied to the center of every grid cell in a pseudorandom order. In a small number of cases, due to operator error, the number of stimuli in a particular grid cell differed from ten, being equal to 9 in 7% of the cells, 11 in 4% and 7, 8 or 12 in less than 1% of the cells. The bootstrapping-based accuracy estimates did not significantly depend on such small variations, which was checked by repeating the calculations using the first eight stimuli in each cell for all the maps. The total number of stimuli in every session was 490. All three sessions were performed with the same intensity equal to 110% of the individual RMT determined in the first session. The coil orientation was tangential to the surface of skull, and the induced electrical field was perpendicular to the central sulcus, in the posterior to anterior direction.

The study was approved by the Ethical Committee of Research Center of Neurology (protocol 9-4/17, 30.08.2017), and written informed consent was obtained from all the participants.

### 2.2. Data Analysis

#### 2.2.1. Muscle Representation Coverage by Grids of Different Sizes

Because the first dataset was acquired without a stimulation grid, the sizes of the obtained representations were not constrained from above and provided a sample from the size distribution in the healthy population. Thus, the maps were used to estimate the fractions of the representations that would be covered by square grids of different sizes centered at the point with the maximum MEP amplitude. Conservative estimates were used, counting only the parts of the representations that were guaranteed to be covered under any grid orientation (i.e., lying within a circle of a radius equal to half the side of the square). The calculations were performed for the following grid sizes: 38, 46, 53, 61 and 69 mm (corresponding to 10, 12, 14, 16 and 18 cells in the Nexstim grid tool) at a peeling depth of 20 mm. The results were compared between the three muscles using the Kruskal–Wallis test.

#### 2.2.2. Muscle Representation Parameters

We calculated the following muscle representation parameters (the formulas are presented in Appendix A):The area of the cells with a mean MEP above 50 µV;The area of the cells with a maximum MEP above 50 µV (or, equivalently, the area of the cells with at least one suprathreshold MEP);The area of the cells with more than half suprathreshold MEPs;The area weighted by the mean MEP amplitude (amplitude-weighted area, also known as map volume [17]);The area weighted by the probability of a suprathreshold MEP (probability-weighted area);The COG with the weights defined as the mean amplitudes in each grid cell;The COG with the weights defined as the maximal amplitudes in each grid cell;The COG with the weights defined as the probabilities of suprathreshold MEPs in each grid cell.

#### 2.2.3. Simulation of Mapping with Different Numbers of Stimuli Using Bootstrapping

To simulate the mapping results that would be obtained with a different number of stimuli per grid cell, we used a bootstrapping-based method [43], in which we randomly chose (with replacement) a given number of values from the 10 amplitudes measured in each cell. The resulting sets of amplitudes were treated as maps, and their parameters were calculated in the same way as for the initial full datasets. Sampling with replacement allows one to simulate arbitrary numbers of stimuli per cell (not necessarily smaller than 10). We performed the calculations for the numbers of stimuli from 1 to 10. The last value corresponds to estimating the accuracy of the representation parameters for our actual protocol. The number of bootstrapping-generated maps was equal to 1000 for every condition.

#### 2.2.4. Bias of the Area and Weighted Area

An important and often overlooked fact is that the accuracy of an estimator is determined not only by its variance but also by the bias, i.e., the difference between the mean value of the estimator and the true value of the estimated parameter. It is necessary to characterize the bias because it can produce spurious effects and make the results obtained using different mapping protocols difficult to compare [44]. The evaluation of the bias is complicated by the inaccessibility of the ‘true values’ of the muscle representation parameters, i.e., those that would be obtained from a hypothetical mapping providing the full knowledge of the MEP probability distributions at every cortical location. 

Our approach to estimating both the bias and variability of representation parameter estimates was based on bootstrapping [45]. Mathematically, the method simulates the mapping results for a muscle representation in which the actual probability distributions of MEP amplitudes in each cell coincide with the empirical distributions obtained in the experiment. It is important, however, that the validity of the estimates does not require exact equality between the empirical and the real MEP distributions, but is based on their approximate similarity, which can be expected with the ten-stimulus sampling.

The normalized (relative) bias was estimated by the following formula:(1)$$B_{norm}\left(P\right)=\frac{mean\left(P\right)-P_{0}}{P_{0}}$$
where P is a muscle representation parameter (such as the area), $P_{0}$ is the parameter value for the experimental map, and mean(P) is the mean parameter value over the maps generated by bootstrapping with a certain number of stimuli per grid cell (ranging from one to ten). 

#### 2.2.5. Within-Session Variability of the Area and Weighted Area

The within-session variability of muscle representation parameters was characterized by the coefficient of variation (CV) of the parameter values for the maps generated by bootstrapping from a given experimental map:(2)$$CV\left(P\right)=\frac{std\left(P\right)}{mean\left(P\right)}$$where std(P) is the sample standard deviation for the bootstrapping-generated maps.

#### 2.2.6. Between-Session Variability of the Area and Weighted Area

The variability of muscle representation parameters between the three mapping sessions was characterized by a variability index equal to one-half of the relative difference of the maximum and minimum values:(3)$$V\left(P\right)=\frac{P_{max}-P_{min}}{P_{max}+P_{min}}$$where $P_{max}$ and $P_{min}$ are maximal and minimal values of the parameter in the three sessions. This quantity measures the relative deviation of these values from their mean. The values of this index were calculated and averaged by 1000 triples of maps generated by bootstrapping from the three mapping sessions.

#### 2.2.7. Sensitivity of the Protocol to Changes between Sessions

The MEP amplitudes in the three mapping sessions were compared in a cell-by-cell manner. Importantly, only the amplitudes above 50 µV could be reliably detected. Thus, the values of all smaller responses were unknown—a situation called ‘data censoring’ in statistics [46]. Accordingly, the samples of MEP amplitudes from every grid cell were compared between the sessions using Gehan’s generalization of the Mann–Whitney test for censored data [47]. To compare the within-session and between-session variability of MEPs, we performed similar tests between the two halves of each sample obtained in a given session (five MEPs in each half for every grid cell). To keep the same statistical power in the between-session tests, we limited them to the first half of each session (five MEPs per grid cell). The test results were visualized using 2D diagrams showing the locations of significant amplitude changes at uncorrected *p* < 0.05. The diagrams are analogous to statistical parametric maps in neuroimaging [48].

To assess whether the representation parameter changes between sessions had the same magnitude as the within-session fluctuations, the parameter distributions for maps generated by bootstrapping from each session were computed. The degree of similarity between pairs of distributions was measured by the overlaps of their histograms, with unit overlap corresponding to identical distributions and zero overlap—to completely incompatible distributions, with no common possible values. If the distributions in two sessions had a small overlap, this was interpreted as a significant change of the parameter between sessions, which could not be explained by the within-session variability.

In addition to the overlap values, it is useful to characterize the parameter heterogeneity in different sessions by a single number. To this end, we calculated the intraclass correlation coefficient (ICC) applied to the three parameter samples generated by bootstrapping from each session. We used the version of the ICC appropriate for the one-way random effects model [49] because the ordering of bootstrapping-generated maps is irrelevant. We call the resulting quantity the *bootstrapping-based between-session intraclass correlation coefficient* (BICC). In a given subject, this index measures the proportion of the parameter variance attributable to systematic session differences. Zero BICC corresponds to a situation in which the changes between sessions can be fully explained by the variability within a session, and high BICC indicates stronger variation between than within sessions. This measure should be distinguished from the ICC applied in the way common in reliability studies, where it is computed for the sets of values obtained in different subjects and quantifies the ability to distinguish the characteristics of different individuals in the presence of variability [49]. Conversely, BICC is calculated for a single subject and measures the ability to discriminate between sessions in the presence of within-session inaccuracy.

#### 2.2.8. Accuracy of the Center of Gravity

The accuracy of the COG was measured by the mean distance between the COG calculated from the experimental map and the COGs of 1000 maps generated by bootstrapping.

## 3. Results

### 3.1. Muscle Representation Coverage by Grids of Different Sizes

For every percentage value X, we calculated the fraction of all healthy subjects for whom at least X per cent of their representation is covered by the grid of a given size (Figure 1). This analysis was performed for the maps from the first dataset obtained without a grid. The coverage fractions were not significantly different between the three muscles (APB, EDC and FDS) for every grid size (*p* > 0.05, Kruskal–Wallis test). Based on this analysis, we selected for the main experiment a grid size of 53 mm (14 cells in the Nexstim grid tool), covering on average 97.9% of the area of the representations. The corresponding mean bias of −2.1% due to incomplete coverage was considered small compared to other factors affecting the area estimates causing variations by up to tens of per cent [50].

### 3.2. Visualization of TMS Maps Obtained with a Stimulation Grid

The mapping results from the grid-based experiment were visualized by representing each grid cell by a square with the color defined by the fraction of the 10 stimuli that produced a suprathreshold MEP (Figure 2A). The muscle representations were generally composed of a region of varying size having a high probability of a suprathreshold response (0.9 and above, colored yellow) and a surrounding area with an intermediate probability (ranging from 0 to 0.9, colored green to dark violet).

### 3.3. Bias of the Area and Weighted Area

The biases of the different variants of the area and weighted area were calculated using the bootstrapping-based map simulation, and their median values from all sessions in all subjects are shown in Figure 3 as functions of the number of stimuli per grid cell. A considerable bias exists in the (unweighted) area parameters, which were defined using thresholding: The area of the cells with the mean MEP above 50 µV, the area of the cells with the maximum MEP above 50 µV and the area of the cells with more than half suprathreshold MEPs. In contrast, the amplitude-weighted area and probability-weighted area had very small biases.

For the area of the cells with more than half suprathreshold MEPs, the bias showed different patterns for even and odd numbers of stimuli per cell, shown separately by the solid and dashed green lines respectively. Moreover, as shown in Appendix C, for particular structures of the representations, the bias of this parameter can be a non-monotonic function of the number of stimuli. The sign of the bias can be negative or positive, depending in a non-trivial way on the details of the representation and the number of stimuli. This suggests interpreting this parameter with caution.

### 3.4. Within-Session Variability of the Area and Weighted Area

The within-session CVs were calculated using the same method as the biases and plotted depending on number of stimuli per grid cell in the maps generated by bootstrapping (Figure 4). For all the parameters, the CV significantly decreased with the number of stimuli per cell (*p* < 0.001, Page’s trend test for ordered alternatives). The parameters having the smallest CVs were the area of the cells with at least one suprathreshold MEP and the probability-weighted area.

The probability-weighted area was characterized by the highest overall accuracy among the considered definitions of the area and weighted area, having a negligible bias and a small CV. This parameter was selected for further analysis of its sensitivity to the between-session map changes (Section 3.6).

### 3.5. Between-Session Variability of the Area and Weighted Area

The between-session variability index demonstrated a pattern similar to that of the within-session CV (Figure 5). The variability index significantly decreases with the number of stimuli per cell (*p* < 0.001, Page’s trend test) for all the parameters except the area of the cells with more than half suprathreshold MEPs, which can have a non-monotonic, subject-dependent bias and should be interpreted with caution (see Appendix C). The parameters with the smallest between-session variability were the area of the cells with at least one suprathreshold MEP and the probability-weighted area (the same parameters that had the smallest within-session CV) as well as the area of the cells with a mean MEP above 50 µV.

### 3.6. Sensitivity of the Protocol to Changes between Sessions

The results of the amplitude comparisons for each grid cell between the first halves of pairs of sessions and between the first and second halves of each session indicate that, on average, the number of significant changes was greater between sessions than within a session (Figure 2B,C).

The relationship between the within-session and between-session variability of the probability-weighted area was characterized by calculating its probability distributions for the maps generated by boostrapping from each session (Figure 6). Five of the eight subjects had a between-session distribution overlap of less than 0.05 in at least one pair of sessions, indicating a significant difference in the probability-weighted area.

We quantified the ability to distinguish the values of the probability-weighted area between sessions using the BICC, i.e., the intraclass correlation coefficient applied to the three parameter samples generated by bootstrapping from each session. The BICC ranged from 0.61 to 0.99. High BICC values (above 0.9) were observed in the three subjects (with numbers 4, 5 and 8) who had zero distribution overlaps in some pairs of sessions. Both measures indicate that in these subjects, the between-session changes of the probability-weighted area were greater than the within-session fluctuations and thus were unlikely to be explainable solely by the trial-to-trial variability of MEPs.

Additionally, to compare the alternative definitions of the area and weighted area by their ability to find significant differences between sessions at the individual level, we computed the BICC values for all the parameters and subjects (Figure 7). In five of the eight subjects, the highest BICC was shown by the probability-weighted area, and in the remaining three subjects – by the amplitude-weighted area.

### 3.7. Accuracy of the Center of Gravity

The COG accuracy was measured by the mean distance between the COG calculated from the initial map with 10 stimuli per cell and the COGs of 1000 maps generated by bootstrapping. The results are shown in Figure 8 depending on number of stimuli per grid cell in the bootstrapping-generated maps. For all the COG variants, this error measure significantly decreased with the number of stimuli (*p* < 0.001, Page’s trend test). The highest accuracy was obtained for the probability-weighted COG, although the accuracy differences with the other two definitions were small (less than 1 mm).

## 4. Discussion

We have studied the impact of the TMS mapping algorithm and data processing on the accuracy of estimating muscle representation parameters. The considered aspects of the mapping procedure were the size of the stimulation grid and the number of stimuli per cell. In regards to data processing, several alternative definitions of the muscle representation area, weighted area and COG were compared in terms of the accuracy of their estimation. Among the considered variants of area and weighted area, the highest overall accuracy was shown by the area weighted by the probability of a suprathreshold MEP. This parameter was further investigated with respect to its sensitivity to the motor map changes between the three sessions recorded on consecutive days. The results show that such changes can be greater than the fluctuations within a session, and thus can be reliably detected in individual subjects using the present protocol. The causes of these changes, including possible physiological and methodological explanations, require further research.

### 4.1. Muscle Representation Coverage by Grids of Different Sizes

The optimal choice of the stimulation grid size has rarely been discussed in the literature. Classen et al. [20] calculated the increasing accuracy of the COGs obtained using square grids with side lengths of 3, 5 and 7 cm. Since the main expected effect of an insufficiently large grid is likely to be the missing of some excitable sites at the periphery, we based our analysis on the percentage of the points with suprathreshold MEPs covered by a grid. The obtained dependence of this characteristic on the grid dimensions can be used to choose an appropriate size that is large enough to ensure the required representation coverage. At the same time, an unnecessarily large grid is undesirable due to the increased mapping time (if the stimulation point density is fixed).

### 4.2. Visualization of TMS Maps Obtained with a Stimulation Grid

The mapping protocol used in this study produced samples of 10 MEP amplitudes from every grid cell in each session. This allowed a statistical comparison of the maps in a cell-by-cell manner—an approach that is widespread in MRI-based neuroimaging, but not so common in TMS mapping (although is occasionally applied [7]). We found a considerable number of significant changes of amplitude distributions between sessions and visualized the spatial configurations of these effects. Significant changes between sessions were more numerous than alterations within a session (i.e., between its first and second halves). This motivates further application of the described methodology for testing location-specific MEP changes with and without interventions based on the MEP samples of considerable size obtained in each grid cell in different mapping sessions.

### 4.3. Bias of the Area and Weighted Area

One of the problems in the field of TMS mapping is the difficulty of comparing results obtained by different groups using a variety of mapping protocols and data processing methods. The performed analysis of the biases of the different variants of area and weighted area indicates that the values of the thresholding-based (unweighted) area definitions have considerable biases. This means that these parameters can systematically differ between protocols with different numbers of stimuli per grid cell. Additionally, every subject is characterized by a particular bias, depending on the details of the MEP probability distributions in all the grid cells (see Appendix B). This means that the influence of the bias cannot be eliminated by a single bias correction procedure. Moreover, if a study applying TMS mapping with a limited number of stimuli compares the representation areas in two groups with systematically different area biases, a totally spurious difference in the area can be obtained. The amplitude-weighted and probability-weighted areas have negligible biases, and thus do not present the above problems.

It should be stressed, however, that the choice of the parameters to focus on in a given study cannot be based solely on their accuracy. Indeed, a parameter may be estimated very accurately, but show no effect in the considered problem. Thus, all muscle representation characteristics can potentially be informative, particularly if their statistical properties are understood and taken into account.

### 4.4. Within-Session Variability of the Area and Weighted Area

The extreme variability of MEP amplitudes (which can span more than two orders of magnitude [51]) leads to the within-session variability of muscle representation parameters [35]. The characteristics considered here involve integration of data from repeated stimulation of many cortical locations, which makes the representation parameters more stable than a single MEP. This effect of stabilization due to averaging was found to vary depending on the exact definition of the representation area or COG. We estimated the variability using a bootstrapping-based method, in which we simulated maps by subsampling MEPs from the datasets recorded in the experiment. 

The results show that the three alternative definitions of the muscle representation area produce different degrees of relative variability (measured by the CV). One of the variants is the area of the cells with at least 6 suprathreshold MEPs (out of 10 stimuli), which was recommended in the protocol proposed in [21] and named “the golden standard” in [44]. This parameter had a larger CV than the other two (unweighted) area variants: the area of the cells with the mean MEP above 50 µV and the area of the cells with at least one suprathreshold MEP. As noted above, this does not imply that any of the parameters should not be used, because although all the three area variants depend on the representation extent, they do not measure exactly the same property and may be sensitive to different effects of interest. The higher variability of the area of the cells with at least 6 suprathreshold MEPs leads to the requirement of larger effect sizes and/or samples for statistical significance as compared to the other two area definitions. Thus, it is important to take into account the accuracy of the different parameters for planning the experiments, even though the accuracy cannot serve as the only basis for parameter selection.

The probability-weighted area has the highest overall accuracy among the area and weighted area variants. It depends on both the extent of the representation and the distributions of MEPs at the included points. Further studies are warranted to assess the utility of this parameter in fundamental and clinical problems.

The obtained decreasing dependencies of errors on the number of stimuli per grid cell can be used for appropriately choosing this number in a given application of TMS mapping. A compromise should be reached between the requirements for high accuracy and reasonable study duration. Several methodological studies of TMS mapping have focused on the number of stimuli sufficient for reliable estimation of representation parameters [20,26,34,36,44,52], and their results may be considered to mean that any further increase in this number is pointless. The obtained dependencies (Figure 4) demonstrate that, although the slopes are largest in the left parts of the curves, the errors continue to decrease for all the considered numbers of stimuli. Thus, a study with a small effect size may benefit from a larger number of stimuli than the minimum one required for reliability.

High parameter accuracy may be especially relevant to investigations of the changes in TMS maps between two time points due to an intervention or spontaneous directed alteration such as disease progression. In such a study, the measured change of a parameter is composed of (1) the constant mean effect of interest, (2) the random change in the mean parameter value between the sessions and (3) the within-session random errors. The error terms (2) and (3) can contain both physiological components (such as excitability fluctuations) and methodological factors (e.g., navigation inaccuracies). The ability to detect the effect (i.e., the statistical power) depends on its size in relation to the error terms (2) and (3). The purpose of sufficient MEP sampling studied in this paper is to reduce the component (3) so that it is small compared to the component (2) and thus does not limit the statistical power. Meanwhile, it is known that the between-session variability (2) is smaller than the between-subject variation, as indicated by the reported ICC values above 0.5 [52,53]. Thus, even a small accuracy gain irrelevant for group comparisons may be essential in pre-post studies.

### 4.5. Between-Session Variability of the Area and Weighted Area

Similarly to the within-session CV, the between-session variability indices of the area and weighted area decreased with the number of stimuli. However, this decrease showed a more pronounced flattening for numbers of stimuli greater than five, in comparison with the decrease of the CV. This is in agreement with the interpretation that increasing the number of stimuli reduces the effect of the short-term MEP variability and brings representation parameters closer to their mean values in a particular session, but these mean values may differ between sessions due to physiological and/or methodological factors. This means that the between-session variability will approach a (nonzero) plateau determined by the differences between the mean values in the sessions. In other words, it is impossible to eliminate the between-session changes by collecting more data in each session.

The relationships between the alternative area definitions were similar to those observed for the within-session CV, with two exceptions. First, the CV was smaller for the amplitude-weighted area than the area of the cells with more than half suprathreshold MEPs, whereas their between-session variability indices were similar. Second, the between-session variability was higher for the probability-weighted area than the area of the cells with at least one suprathreshold MEP, whereas their CVs were similar. This may correspond to a greater day-to-day stability of the representation ‘footprint’ (the area of the region able to produce MEPs) than its ‘height’ in terms of MEP probability (i.e., the average degree of certainty with which a suprathreshold response will be elicited in each location). 

### 4.6. Sensitivity of the Protocol to Changes between Sessions

An optimal TMS mapping protocol for a given study must be sensitive to the effect being investigated. As mentioned above, an important type of research question concerns the changes of TMS maps with time, e.g., in the course of disease progression [14] or as a result of neuroplasticity caused by therapeutic interventions [15]. To reliably detect such changes at the level of individual subjects, the within-session variability should be small compared to the between-session effect size. In the present study, we compared three mapping sessions without any interventions between them. The changes in the probability-weighted area between the consecutive days were shown to be greater than the within-session fluctuations. This suggests that the day-to-day changes in this parameter cannot be fully explained by the inaccuracy produced by the trial-to-trial MEP amplitude variability.

### 4.7. Accuracy of the Center of Gravity

Similarly to the CV of the extent-related representation parameters, the within-session errors in the center of gravity decreased with the number of stimuli. They were within the nominal accuracy of the navigation system (5.7 mm). The probability-weighted COG showed a slightly higher accuracy than the COGs weighted by the mean and maximum MEPs, which may be due to its independence of large fluctuations in the MEP amplitudes known to have a heavy-tailed distribution [51].

### 4.8. Limitations

A limitation of the study is the small number of subjects. However, it can be argued that for the bootstrapping-based accuracy assessment, the total amount of MEP data is of primary importance, and this amount was substantial due to the detailed mapping protocol.

It is important to note that the results of the study were obtained in healthy individuals, and the analysis of their applicability to patient populations requires additional studies. 

## 5. Conclusions

We have studied the dependence of the accuracy of muscle representation parameters on the aspects of the grid-based TMS mapping experiment and data processing. The grid size impacted the completeness of the muscle representation coverage, and a square grid with a side of 53 mm (at the peeling depth of 20 mm) centered at the hotspot covered on average 97.9% of the representation area for the APB, EDC and FDS muscles. The within-session accuracy of the representation area, weighted area and COG improved with the increasing number of stimuli without saturation up to at least ten stimuli per cell. For the area definitions based on thresholding, a considerable bias was observed for small numbers of stimuli, while for the probability-weighted and mean amplitude-weighted areas the bias was negligible. The area weighted by the probability of a suprathreshold MEP showed the highest overall accuracy among the considered definitions of the area and weighted area (surpassing the accuracy of the commonly considered area of the cells with more than half suprathreshold MEPs). The protocol was found to have sufficient sensitivity to distinguish the between-session changes of the probability-weighted area from its within-session fluctuations. The results can guide the choice of the grid size, the number of stimuli per cell and the investigated representation parameters in studies applying TMS mapping to research and clinical problems.

## Figures and Tables

**Figure 1 brainsci-09-00088-f001:**
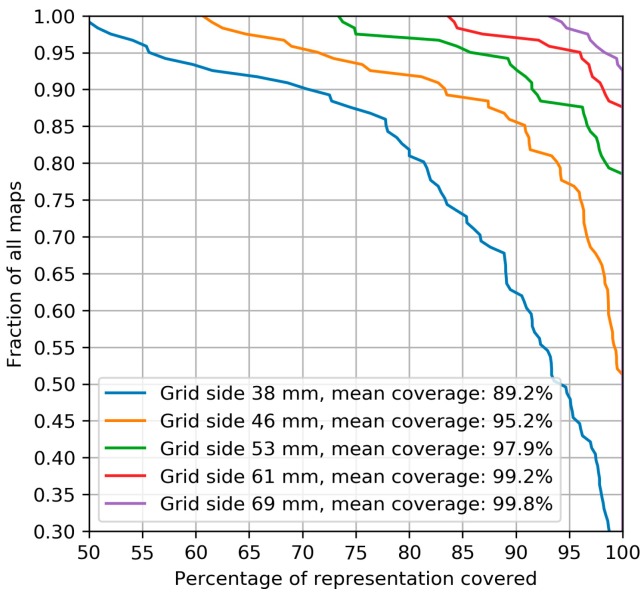
The effect of the grid size on the completeness of muscle representation coverage. For every percentage value X, the corresponding Y value is the fraction of all the maps in which at least X per cent of the representation is covered by the grid of a given size. The simulated grids were located at the peeling depth of 20 mm, centered at the point with the maximum motor evoked potential (MEP) amplitude, had a square shape, and their side lengths were chosen as even integer multiples of a cell in the Nexstim grid tool (10 to 18 cells). The transcranial magnetic stimulation (TMS) maps used in this calculation were obtained without a grid in healthy subjects (13 maps for abductor pollicis brevis (APB), 54 for extensor digitorum communis (EDC), and 54 for flexor digitorum superficialis (FDS)). There was no significant difference in the representation coverage between the muscles for every grid size (*p* > 0.05, Kruskal-Wallis test). Note: The estimates of the coverage are conservative in that stimulation points were counted as covered by the grid only if their distance from the hotspot was smaller than one-half of the side of the grid (i.e., excluding the coverage by the corners of the square, which is possible, but not guaranteed under varying grid orientations). The green line corresponds to the grid size used in the present study.

**Figure 2 brainsci-09-00088-f002:**
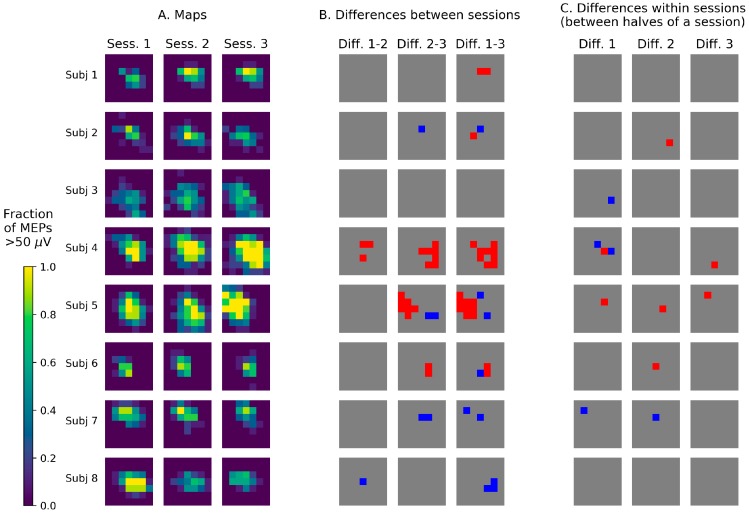
(**A**) TMS maps of the APB muscle from three sessions performed on consecutive days. The squares represent the stimulation grid cells, and the color encodes the fraction of the applied 10 stimuli that produced suprathreshold MEPs (above 50 µV). (**B**) Results of the comparison of the first five MEP amplitudes in each cell between sessions using Gehan’s generalization of the Mann–Whitney test for censored data. The red cells had significantly greater amplitudes in the second session of the compared pair (with uncorrected *p* < 0.05), and the blue cells—significantly smaller amplitudes. (**C**) Results of the comparison of the first and the second five MEP amplitudes in each cell. The test and the color code are the same as in B.

**Figure 3 brainsci-09-00088-f003:**
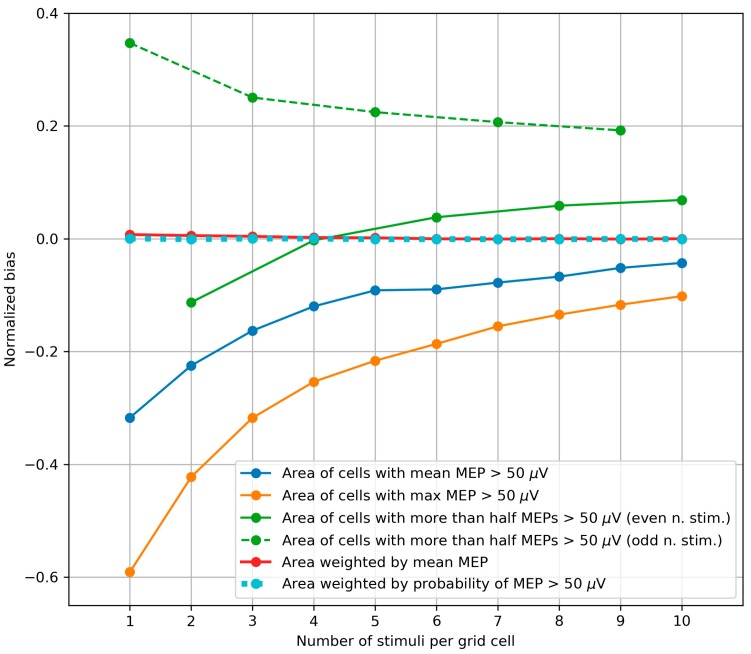
The dependence of the normalized bias of the representation parameters on the number of TMS stimuli per grid cell. The values of the parameters were averaged by 1000 maps generated by bootstrapping (with replacement) from every experimental map obtained with 10 stimuli per cell in each subject. The median values from all maps of all subjects are depicted. The area of the cells with more than half suprathreshold MEPs showed different patterns for even and odd numbers of stimuli per cell (see also Figure A3 in Appendix B). These subseries are shown separately by the solid and dashed green lines respectively (here and in Figure 4 and Figure 5). The biases of the mean amplitude-weighted and probability-weighted areas are close to zero, and thus the red and cyan curves are close to the X axis.

**Figure 4 brainsci-09-00088-f004:**
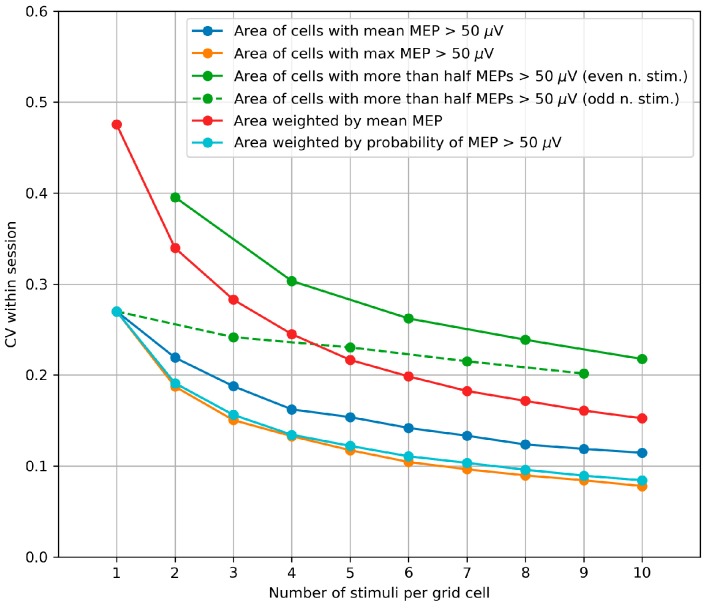
Within-session variability of area parameters measured by the coefficient of variation (CV) of the parameter values obtained in 1000 maps generated by bootstrapping from every initial 10-stimuli-per-cell map of every subject. The median values of the CVs from all maps of all subjects are depicted. For all the parameters, the CV significantly decreases with the number of stimuli per cell (*p* < 0.001, Page’s trend test for ordered alternatives).

**Figure 5 brainsci-09-00088-f005:**
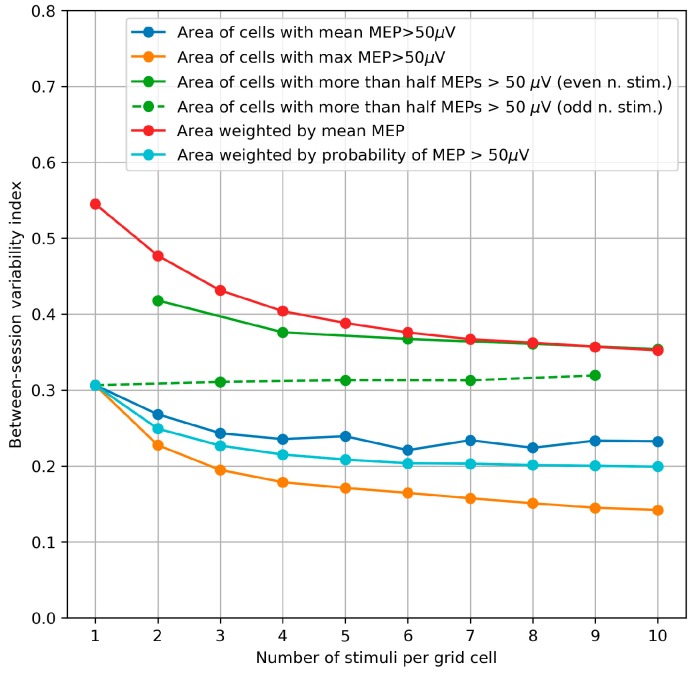
Between-session variability of the area parameters measured by an index equal to one-half of the relative difference of the maximum and minimum values among the three mapping sessions performed on consecutive days. These indices were calculated and averaged by 1000 triples of maps generated by bootstrapping from the MEPs obtained in the three sessions. The median values from all subjects are shown in the plot. The variability index significantly decreases with the number of stimuli per cell (*p* < 0.001, Page’s trend test) for all the parameters except the area of the cells with more than half suprathreshold MEPs, which can have a non-monotonic, subject-dependent bias and should be interpreted with caution (see Appendix C).

**Figure 6 brainsci-09-00088-f006:**
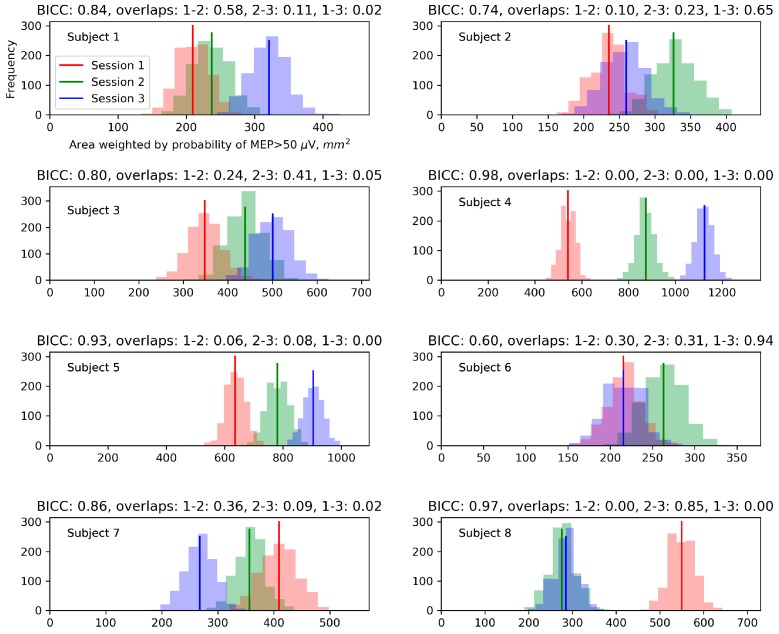
Comparison of the between-session and within-session variability for the probability-weighted area (sum of grid cell areas multiplied by the probabilities of suprathreshold MEPs in them). Each plot corresponds to one subject and shows three histograms for different mapping sessions. Each histogram shows the within-session distribution of the values of the probability-weighted area obtained from 1000 maps generated by bootstrapping from a given map. Above the plots, the measures of the possibility to discriminate between sessions are shown: the bootstrapping-based between-session intraclass correlation coefficient (BICC) and the pairwise distribution overlaps.

**Figure 7 brainsci-09-00088-f007:**
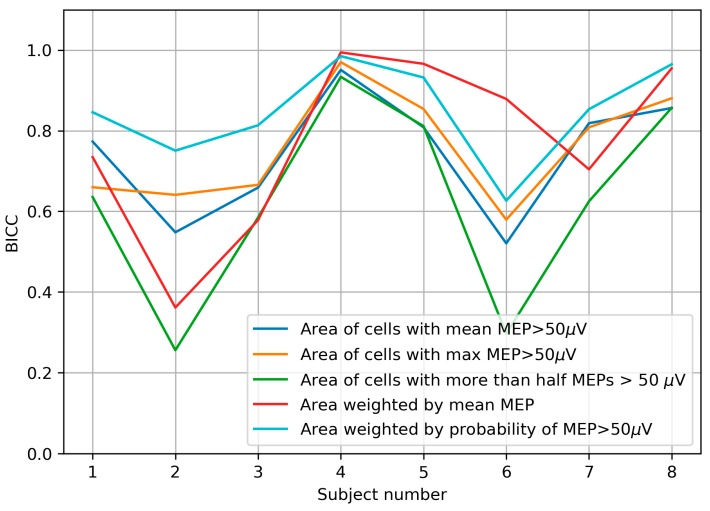
Bootstrapping-based between-session intraclass correlation coefficient (BICC) calculated for all the area and weighted area variants in all subjects.

**Figure 8 brainsci-09-00088-f008:**
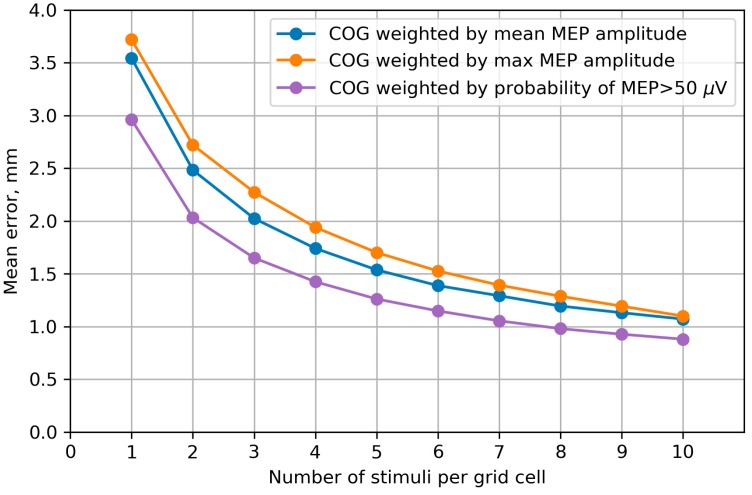
Accuracy of the center of gravity (COG) estimates computed using three alternative methods of assigning weights to the stimulation points. The accuracy is measured by the mean distance between the COG calculated from the full map and the COGs of 1000 maps generated by bootstrapping. The median values from all maps of all subjects are shown. The differences in the COG accuracy between the three methods are statistically significant for all numbers of stimuli smaller than 8 (*p* < 0.05, Friedman test). The highest accuracy is achieved by the approach in which the stimulus location vectors are weighted by the probability of a suprathreshold MEP in them (purple curve), although the accuracy differences between the methods are small (less than 1 mm). For all the COG variants, the error significantly decreases with the number of stimuli per cell (*p* < 0.001, Page’s trend test).

## Data Availability

The TMS mapping data and the source code of the scripts used for data processing are available at https://github.com/DOSinitsyn/gridTMSmaps.

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
