# Peer review of "Optimization of the Navigated TMS Mapping Algorithm for Accurate Estimation of Cortical Muscle Representation Characteristics"

_brainsci, 2019, doi:10.3390/brainsci9040088_

Reviewer 1 Report

In this study, Sinitsyn and colleagues aimed to optimize the TMS mapping by means of an algorithm based on the accuracy of muscle representation parameters. For this aims, two experiments were performed. In the first study, TMS maps over three different muscles (APB, EDC and FDS) of 33 subjects were recorded to determine an optimal size of the stimulation point grid for the second experiment. In the second experiment, the cortical representations of the APB muscle of 8 subjects were mapped three times on consecutive days.

The study is very interesting and well written. From a methodological point of view, it is innovative and strongly relevant for the field of research in which a navigated TMS approach is used. I would specify that due to my profile, I provide my evaluation by focusing on the TMS part of the study, as I am not an expert in data processing algorithms. At this regard, I have some suggestions that the authors could take in consideration for ameliorate their manuscript.

In the title, the terms plasticity and abnormalities result ambiguous considering that this study does not include an evaluation of navigated TMS mapping of cortical muscle representations before and after specific interventions or in neurological diseases.

A most clear explanation of the rationale underlying the two studies should be included, as well as the link between them. More specifically, a justification about the choice to perform the TMS mapping only for APB muscle and on just on 8 subjects should be reported (as performed in the second and main experiment). Additionally, why the authors chose a grid size of 53 mm covering on average 97.9% of the area of the representations?

Regarding the TMS delivering, what was the rational of using a biphasic TMS pulse? A more precise description of the utilized TMS pulse waveform and of the current-direction (e.g. a biphasic pulse inducing a posterior-anterior followed by an anterior-posterior current flow in the brain) would improve the description of the method used.

Considering that the number of TMS pulses commonly used in literature is about 20 stimuli, could the authors explain why only 10 stimuli were applied over the grid cells?

What was the inter stimulus interval between each TMS pulse? The authors should consider the recent relevant studies about this point (e.g., Julkunen et al. Does second-scale intertrial interval affect motor evoked potentials induced by single-pulse transcranial magnetic stimulation? Brain Stimul 2012;5:526–32; Pellicciari et al. Ongoing cumulative effects of single TMS pulses on corticospinal excitability: An intra- and inter-block investigation. Clin Neurophysiol. 2016 Jan;127(1):621-628).

A table with the individual resting motor threshold of the eight healthy subjects for each experimental session should be reported.

Finally, I share with the authors that the major limitation of this study is the small number of subjects. I suggest to increase the sample size to consolidate the relevant findings of this study.

Author Response

Reviewer 1

We thank the reviewer for the careful reading of the manuscript and thoughtful comments. We believe they helped improve the quality of the paper.

In the title, the terms plasticity and abnormalities result ambiguous considering that this study does not include an evaluation of navigated TMS mapping of cortical muscle representations before and after specific interventions or in neurological diseases.

Following this comment, we have changed the title to:

“Optimization of the navigated TMS mapping algorithm for accurate estimation of cortical muscle representation characteristics.”

A most clear explanation of the rationale underlying the two studies should be included, as well as the link between them. More specifically, a justification about the choice to perform the TMS mapping only for APB muscle and on just on 8 subjects should be reported (as performed in the second and main experiment).

The rationale underlying the first study is described in the Methods section as follows:

“The first dataset was recorded previously for different purposes. It was used here to determine an optimal size of the stimulation point grid for the second (main) experiment.”

As regards the second study, we have added the following explanation:

“The second (main) experiment aimed at investigating the relationship between the MEP sampling scheme in the grid-based mapping algorithm and the representation parameter accuracy.”

We have also included a passage about the choice of the APB muscle:

“The APB muscle was chosen because of its frequent use in TMS mapping due to its relatively large cortical representations and low baseline EMG activity [40].”

The mapping of a single muscle was motivated by time considerations, as our protocol is relatively time-consuming (490 stimuli per map). Concerning the number of subjects, as mentioned in the Limitations section, we believe that for the bootstrapping-based accuracy assessment, the total amount of MEP data from all subjects is of primary importance, and this amount was substantial due to the detailed mapping protocol.

Additionally, why the authors chose a grid size of 53 mm covering on average 97.9% of the area of the representations?

We note that the sizes and shapes of the representations are variable between subjects, and so it is challenging to choose a universally optimal grid shape and size. A small grid can introduce a negative area bias, whereas an unnecessarily large grid will increase the mapping time (for a fixed stimulus density). As regards the size chosen in the present study, we have added the following explanation:

“The corresponding mean bias of -2.1% due to incomplete coverage was considered small compared to other factors affecting the area estimates causing variations by up to tens of per cent [47].”

Regarding the TMS delivering, what was the rational of using a biphasic TMS pulse?

The following sentence has been included:

“Biphasic pulses were used because they yield the lowest motor thresholds [39,40], which mitigates the problem of coil heating.”

A more precise description of the utilized TMS pulse waveform and of the current-direction (e.g. a biphasic pulse inducing a posterior-anterior followed by an anterior-posterior current flow in the brain) would improve the description of the method used.

We have rewritten the description of the TMS pulse waveform in the following way:

“We used a figure-of-eight biphasic coil with a diameter of 50 mm to deliver stimuli with a 280 μs duration. The pulses induced a posterior-anterior followed by an anterior-posterior current flow in the brain.”

Considering that the number of TMS pulses commonly used in literature is about 20 stimuli, could the authors explain why only 10 stimuli were applied over the grid cells?

We note that in the majority of the studies we are aware of, the total number of stimuli in the grid-based mapping procedure (all cells combined) varied from 147 [Van De Ruit, 2015] to 250 [Weiss, 2013]. Only in a single paper from 1998 did we find a protocol with 20 stimuli per cell, with 7x7 cells in the grid [Classen, 1998]. Our experimental algorithm involved 10 stimuli applied to each of 7x7=49 stimulation sites, which brings the total number of stimuli to 490. This number is relatively large compared to recent studies, and a more extended protocol would have a disadvantage of consuming a lot of time and inducing subject fatigue.

Van De Ruit, M.; Perenboom, M.J.L.; Grey, M.J. TMS brain mapping in less than two minutes. Brain Stimul. 2015, 8, 231–239.

Weiss, C.; Nettekoven, C.; Rehme, A.K.; Neuschmelting, V.; Eisenbeis, A.; Goldbrunner, R.; Grefkes, C. Mapping the hand, foot and face representations in the primary motor cortex - Retest reliability of neuronavigated TMS versus functional MRI. Neuroimage 2013, 66, 531–542.

Classen, J.; Knorr, U.; Werhahn, K.J.; Schlaug, G.; Kunesch, E.; Cohen, L.G.; Seitz, R.J.; Benecke, R. Multimodal output mapping of human central motor representation on different spatial scales. J. Physiol. 1998, 512 ( Pt 1), 163–79.

What was the inter stimulus interval between each TMS pulse? The authors should consider the recent relevant studies about this point (e.g., Julkunen et al. Does second-scale intertrial interval affect motor evoked potentials induced by single-pulse transcranial magnetic stimulation? Brain Stimul 2012;5:526–32; Pellicciari et al. Ongoing cumulative effects of single TMS pulses on corticospinal excitability: An intra- and inter-block investigation. Clin Neurophysiol. 2016 Jan;127(1):621-628).

We have added the following sentence: “The inter-stimulus interval was greater than two seconds”.

We note that this interval was determined by the time needed to move the coil from one grid cell to the next. Julkunen et al. (2012) conducted their main analysis for 31-fold repeated stimulation of the same location. They report heterogeneous subject-specific effects of the inter-stimulus interval and suggest that “random protocols that mix sub- and suprathreshold TMS appear as more stable for individual subjects”. The present study is concerned with mapping, and the location is changed after every stimulus. Thus, in addition to making fewer (ten) stimulations of the same cortical site than in [Julkunen et al., 2012], we distributed them in time, alternating them with the stimulation of other locations, which bears some resemblance with the above-mentioned mixed approach.

A table with the individual resting motor threshold of the eight healthy subjects for each experimental session should be reported.

We have included the requested table - Table 1 in Appendix D.

 Finally, I share with the authors that the major limitation of this study is the small number of subjects. I suggest to increase the sample size to consolidate the relevant findings of this study.

We thank the reviewer for this comment, and we are planning to further apply the present algorithm to healthy volunteers in future research to validate our conclusions.

Reviewer 2 Report

Optimization of the navigated TMS mapping algorithm for accurate detection of pasticity and abnormalities in cortical muscle representation.

Sinitsyn et al., brainsci-476137

     The authors set out to test mapping procedures with navigated TMS to establish consistency in measures and sessions. They targeted three common muscle groups with 110% of RMT stimulation in a small number of participants over a couple sessions. Their findings suggest some benefits for their methods, in certain circumstances and instantiate the variability of RMT overall. The general study outline and execution is solid in attempting to answer an important question. A few clarifications could benefit the overall manuscript.

     Could the authors please describe in more detail:

-       How they identified the hot spot for each participant.

-       How the T1 weighted MRI was used for navigation

-       Which motor cortex was targeted (was it for the dominant hand such that it was the right hemisphere stimulated for the left-handed participants? If not, perhaps the left-handed participants should be excluded)

-       What was the coil orientation on the scalp? Generally, people report a 45 degree tilt to the coil.

-       What was the inter-stimulation interval? It’s important that sufficient time between stimulations is given for the cortex to relax back to baseline.

-       Were muscle movements, beyond MEP amplitudes, used in any way to identify RMT? If not, why not? It is possible to induce movement in the target, or non-target, muscle without a MEP of amplitude to meet criterion.

In the figures where odd and even stimulations are plotted, the X-axis labels do not match the data plotted. For the two green lines, it plots stimulation 1, 3, 5, or 2, 4, 6, etc., but in the other colored lines it plots actual numbers of stimuli selected.

In figure 3, it is unclear thy the blue and orange lines are included. Also, the legend includes a red line which is not plotted.

Three muscle groups were targeted but no comparisons among the groups is included here. Why not?

The variability between sessions is an important aspect of this work and I believe to be substantial contribution to the field.

Author Response

Reviewer 2

We thank the reviewer for the careful reading of the manuscript and thoughtful comments. We believe they helped improve the quality of the paper.

        Could the authors please describe in more detail:

-       How they identified the hot spot for each participant.

We have added the following paragraph:

“The hot spot was identified by stimulating the hand knob and the adjacent areas by no less than 20 stimuli at an intensity sufficient for inducing MEPs with amplitudes of 100-500 μV. The point with the maximal MEP amplitude was considered the hot spot.”

-       How the T1 weighted MRI was used for navigation

We have added the sentence:

“A 3D brain surface was reconstructed based on the T1-weighted structural MR images and used in the Nexstim navigated TMS software for visualizing the brain morphology during mapping.”

-       Which motor cortex was targeted (was it for the dominant hand such that it was the right hemisphere stimulated for the left-handed participants? If not, perhaps the left-handed participants should be excluded)

We have added that the mapped muscles were dominant in the first experiment. As regards the second experiment, the description contains the corresponding information: “The cortical representations of the right APB muscle in 8 healthy volunteers (3 women, median age 28, age quartiles 24, 29, all right-handed…”

-       What was the coil orientation on the scalp? Generally, people report a 45 degree tilt to the coil.

The orientation is described in the following sentence:

“The coil orientation was tangential to the surface of skull, and the induced electrical field was perpendicular to the central sulcus, in the posterior to anterior direction.”

This results in an orientation that depends on the individual local sulcus morphology, but is approximately 45 degrees to the midsagittal line.

-       What was the inter-stimulation interval? It’s important that sufficient time between stimulations is given for the cortex to relax back to baseline.

We have added the following sentence: “The inter-stimulus interval was greater than two seconds”.

This interval was determined by the time needed to move the coil from one grid cell to the next. We note that the location was changed after every stimulus, and thus, the time between the stimulations of the same site was equal to several inter-stimulus intervals (determined by the pseudorandom order of the stimulation points).

-       Were muscle movements, beyond MEP amplitudes, used in any way to identify RMT? If not, why not? It is possible to induce movement in the target, or non-target, muscle without a MEP of amplitude to meet criterion.

We used the MEP amplitude since it can be objectively measured, whereas movements currently require subjective analysis.

In the figures where odd and even stimulations are plotted, the X-axis labels do not match the data plotted. For the two green lines, it plots stimulation 1, 3, 5, or 2, 4, 6, etc., but in the other colored lines it plots actual numbers of stimuli selected.

We note that the green curves represent the same series of values separated into even and odd subseries. We have modified the explanation of this in the caption to Fig. 3 in the following way:

“The area of the cells with more than half suprathreshold MEPs showed different patterns for even and odd numbers of stimuli per cell (see also Fig. 11 in Appendix B). These subseries are shown separately by the solid and dashed green lines respectively (here and in Figs. 4, 5).”

The other colored lines each correspond to a whole series of values, without separation into odd and even argument values, because these parameters did not show any prominent effect of the parity (evenness or oddness) of the number of stimuli.

In figure 3, it is unclear thy the blue and orange lines are included. Also, the legend includes a red line which is not plotted.

The lines of different colors in Figs. 3-5 correspond to different motor representation parameters, i.e. the area defined in several alternative ways and two types of weighted area. Thus, the blue and orange lines are included because they depict the bias for the first two area definitions (the area of the cells with the mean MEP amplitude above 50 μV and the area of the cells with the maximal MEP amplitude above 50 μV, as stated in the figure legend).

We have remade this figure: the former purple line has now been plotted cyan (in this and the following two figures) and its style has been changed to dotted, so that both this line and the red one are now visible. These lines are both close to the X axis, which is explained in the caption as follows: “The biases of the mean amplitude-weighted and probability-weighted areas are close to zero, and thus the red and cyan curves are close to the X axis.”

Three muscle groups were targeted but no comparisons among the groups is included here. Why not?

There is a sentence with the results of the comparison of the muscles in terms of the representation coverage by the simulated grids of different sizes:

 “The coverage fractions were not significantly different between the three muscles (APB, EDC and FDS) for every grid size (p>0.05, Kruskal-Wallis test).“

This aspect was the only one analyzed in the first experiment, because the goal of this analysis was the appropriate choice of the grid size for the second (main) experiment. The second experiment was performed only for the APB muscle (the algorithm used is time-consuming even for a single muscle).

We thank the reviewer for drawing our attention to this issue because it helped us find and fix a typo in the caption to Fig. 1: we have changed ‘representation size’ for ‘representation coverage’ in the description of the between-muscle comparison.

Round  2

Reviewer 1 Report

 In this revised manuscript the authors have satisfactorily addressed all my comments.